# Telomerase Inhibitor TMPyP4 Alters Adhesion and Migration of Breast-Cancer Cells MCF7 and MDA-MB-231

**DOI:** 10.3390/ijms20112670

**Published:** 2019-05-30

**Authors:** Natalia Konieczna, Aleksandra Romaniuk-Drapała, Natalia Lisiak, Ewa Totoń, Anna Paszel-Jaworska, Mariusz Kaczmarek, Błażej Rubiś

**Affiliations:** 1Department of Clinical Chemistry and Molecular Diagnostics, Poznan University of Medical Sciences, 49 Przybyszewskiego St., 60-355 Poznań, Poland; koniecznanatalia2@gmail.com (N.K.); aromaniuk@ump.edu.pl (A.R.-D.); nlisiak@ump.edu.pl (N.L.); etoton@ump.edu.pl (E.T.); apaszel@ump.edu.pl (A.P.-J.); 2Department of Medical Diagnostics, 38A Dobra St., 60-595 Poznań, Poland; 3Department of Immunology, Chair of Clinical Immunology, Poznan University of Medical Sciences, 5D Rokietnicka St., 60-806 Poznań, Poland; markacz@ump.edu.pl

**Keywords:** TMPyP4, telomerase, hTERT, adhesion, migration, cell cycle, breast cancer, MCF7, MDA-MB-231, MCF-12A

## Abstract

Human telomeres were one of the first discovered and characterized sequences forming quadruplex structures. Association of these structures with oncogenic and tumor suppressor proteins suggests their important role in cancer development and therapy efficacy. Since cationic porphyrin TMPyP4 is known as G-quadruplex stabilizer and telomerase inhibitor, the aim of the study was to analyze the anticancer properties of this compound in two different human breast-cancer MCF7 and MDA-MB-231 cell lines. The cytotoxicity of TMPyP4 alone or in combination with doxorubicin was measured by MTT (3-[4,5-dimethylthiazol-2-yl]-2,5-diphenyl tetrazolium bromid) and clonogenic assays, and the cell-cycle alterations were analyzed by flow cytometry. Telomerase expression and activity were evaluated using qPCR and telomeric repeat amplification protocol (TRAP) assays, respectively. The contribution of G-quadruplex inhibitor to protein pathways engaged in cell survival, DNA repair, adhesion, and migration was performed using immunodetection. Scratch assay and functional assessment of migration and cell adhesion were also performed. Consequently, it was revealed that in the short term, TMPyP4 neither revealed cytotoxic effect nor sensitized MCF7 and MDA-MB-231 to doxorubicin, but altered breast-cancer cell adhesion and migration. It suggests that TMPyP4 might substantially contribute to a significant decrease in cancer cell dissemination and, consequently, cancer cell survival reduction. Importantly, this effect might not be associated with telomeres or telomerase.

## 1. Introduction

Restoration of telomerase expression and activity in cancer cells is associated with telomere stabilization and is essential for cell immortalization and tumorigenicity [1]. Consequently, telomerase has become a good candidate in cancer elimination approach. It was reported that telomerase down-regulation provoked cancer cell elimination via two different ways—telomere length-dependent (effect of telomerase inhibition) and independent (mediated through hTERT repression or translocation and some non-canonical effects) [2]. Both strategies may lead to a decrease in tumor cell proliferation, replicative senescence, and activation of apoptosis [3,4].

One of the telomere-associated anticancer approaches is based on blocking telomerase access to the telomeres by forming and stabilizing the structure of the G-quadruplex at telomeric ends. The telomeric end fragment (150–200 nucleotides) usually forms a single-stranded structure and is naturally prevented from the formation of higher-order structures by the presence of proteins that safeguard telomeres. These are, in particular, POT1 (protection of telomeres 1) and TPP1 (TIN2 and POT1-interacting protein 1). Dysfunction of the proteins leads to a loss of telomere protection and induction of DNA damage-activated response and cell proliferation arrest [5]. Interestingly, small molecules that can compete with these proteins induce a cellular response [6]. It is noteworthy that folding the single-stranded telomeric DNA (telomerase’s substrate) into a four-stranded quadruplex structure inhibits the enzyme’s catalytic activity. Among the ligands that can stabilize the G-quadruplex structure with the highest efficiency and specificity in cancer cells, telomestatin, RHPS4, BRACO-19, and TMPyP4 can be found [7].

At the later stages of cancer development, the main cause of cancer-related death is not only the impaired proliferation of cancer cells but also compromised ability of cancer cells to migrate and form metastases. Adhesion and migration properties of cancer cells change during the stages of cancerogenesis. This may be both beneficial as well as disadvantageous for tumor development. On the one hand, loss of adhesion properties initiates cancer cell detachment and invasion into surrounding tissues. Consequently, it may lead to migration out of primary tumors and metastasis [8,9]. However, targeting adhesion molecules and inhibition of the cell–cell communication system may be an efficient way to suppress metastasis and control cancer development. This strategy has been shown to be profitable in many animal models and is still being developed with the use of monoclonal antibodies, synthetic peptides, soluble adhesion molecules, antisense oligonucleotides, etc. [10,11].

Regulation of adhesion and migration of cancer cells is complex and includes both interactions between proteins (e.g., cell–cell adhesion molecules, cadherins, integrins, MAPKs, receptor tyrosine kinases, cytoskeleton proteins) and modulation of different signaling pathways that are known to control cell proliferation, survival, and differentiation [12]. Recent studies demonstrated that the catalytic component of human telomerase—hERT (human telomerase reverse transcriptase)—revealed some non-canonical functions beyond catalytic activity. Specifically, hTERT-associated pathways contribute to apoptosis regulation, DNA damage response, and transcriptional regulation of the Wnt and NF-kB signaling pathways [2]. There are also some reports suggesting that hTERT might promote cell adhesion and migration by affecting the expression of cell adhesion-related genes [13]. However, the exact pathway mediating those mechanisms is still unknown.

Some ongoing clinical trials are aiming to assess the effects of targeting cancer cell adhesion and/or migration. There are also some other approaches that are dynamically developing towards telomerase-targeting strategies. In addition, even if they are both showing some spectacular effects, they did not bring satisfying results in cancer treatment. Thus, we come to a natural conclusion that the only way to fight cancer is to link up different approaches through acting on multiple stages of cancer development.

In this report, we used the cationic porphyrin TMPyP4 to inhibit telomerase in human breast-cancer cell lines and evaluated the mechanism of action of the porphyrin. Presumably, TMPyP4 action is not limited to the influence on telomerase by G-quadruplex stabilization. As demonstrated, photodynamic therapy with TMPyP4 led to the formation of reactive oxygen species and changes in expression of genes involved in oxidative stress response [14]. Moreover, some reports suggest the complexity of TMPyP4 action in cancer cells including not only G4 stabilization but also a contribution to the regulation of expression of some genes engaged in cell metabolism, proliferation, and survival [15,16]. Consequently, some intense studies are still being conducted to reveal how compounds with such biological potential might be used in cancer therapy, especially since they might provoke biological effects in a telomere-independent way.

## 2. Results

### 2.1. Cellular Toxicity of TMPyP4

The experiments were performed with the use of telomerase-positive breast-cancer cells, MCF7 and MDA-MB-231. They are often used for the assessment of cancer therapy based on doxorubicin, and both belong to the NCI-60 cancer cell line panel used by the National Cancer Institute (NCI) for the screening of compounds to detect potential anticancer activity. Importantly, MCF7 cells are caspase-3 deficient that determines the resistance of those cells to some cytotoxic agents [17]. Additionally, a non-tumorigenic epithelial cell line MCF12A was also examined to investigate a potential cancer cell-specific response.

In the MTT studies, we found that all cell lines responded similarly to the TMPyP4 when a 24 h treatment was applied (Figure 1). However, the effect was dose- and time-dependent. Interestingly, during 48 h incubation, cell survival was diminished significantly, but it never exceeded a 30% decrease (100 µM TMPyP4, relative to control cells). The cell viability reduction was more efficient when 72 h incubation was applied, showing up to about 75% reduction in cancer cells and no more than 40% decrease in the non-cancer cell line. Thus, some specificity towards cancer cell targeting was observed.

Interestingly, co-treatment of studied cells with the porphyrin and doxorubicin (DOX) did not show any significant additive effect. We could only see the dominant effect of DOX. That indicates no effect of TMPyP4 on sensitization to DNA-damaging drug in those specific experiments’ conditions (Figure 1). It is worth noting that DOX concentration, i.e., 0.1 µM, was chosen based on the MTT assay (Appendix A). We selected the concentration that provoked the lowest significant but reproducible toxicity to avoid too high concentration that might reveal non-specific effects.

### 2.2. TMPyP4 Alters Telomerase Expression and Activity

Since MCF-12A cells were reported as non-tumorigenic with residual telomerase expression/activity [18], further analysis was performed with the use of cancer cell lines only. Consequently, we decided to verify the potential of TMPyP4 to modulate telomerase and we observed a significant decrease of the key telomerase subunit expression in both MCF7 (Figure 2A) as well as MDA-MB-231 cells (Figure 2B). It is worth noting that the effect was much more significant in MCF7 cells where the 10 µM TMPyP4 provoked a 50% decrease while 20 and 50 µM TMPyP4 caused around 90% hTERT down-regulation, respectively. In MDA-MB-231 cells, the effect was not as profound, and 10 µM porphyrin did not affect hTERT expression while the other two concentrations down-regulated hTERT by ca 40% when applied alone (Figure 2B). Interestingly, we also observed a dramatic fall of hTERT expression after low concentration of DOX (0.1 µM) for 72 h in MCF7 (Figure 2A). Consequently, it was impossible to see any cumulative effect of both compounds if both disrupted hTERT expression so radically. Alternatively, in MDA-MB-231 cells, doxorubicin did not cause any significant down-regulation of hTERT expression, but it did not either provoke an increase in the TMPyP4-mediated down-regulation effect. Very similar effects were observed when telomerase activity was evaluated. In MCF7 cells, treatment with TMPyP4 in all concentrations (i.e., 10, 20, or 50 µM), DOX alone (0.1 µM) or combination of those two compounds provoked a significant (more than 80% in all samples) decrease of the enzyme activity (Figure 2C). MDA-MB-231 cells once again appeared to be slightly more resistant to the test compounds. When cells were treated with 10 µM TMPyP4, the telomerase activity decreased by ca 50% and treatment with higher concentrations, DOX alone, or a combination of these compounds led to a radical decrease in the enzyme activity (more than 80% inhibition) (Figure 2D). It is worth noting that MCF7 cells showed a significantly higher basal level of telomerase catalytic subunit than MDA-MB-231 cells (Figure 2E,F). Since there was no significant difference between those two lines in MTT assay, this suggested that telomeres and hTERT may not be the only target for TMPyP4.

To verify the specificity of the effect we performed the analysis of hTERT protein level in cancer cells after exposure to TMPyP4, BIBR 1532 (both at 20 µM) or DOX (0.1 µM) in a time-course experiment. We used BIBR 1532 to verify any potential similarity in the action of the two telomerase inhibitors effect, but it appeared that this compound was not that efficient either in hTERT down-regulation (Figure 3A,B) considering time intervals of 24, 48, or 72 h.

We observed a time-dependent and significant down-regulation of hTERT after treatment with porphyrin in both cell lines. However, the effect was more efficient in MCF7 cells showing a very low level of the protein after 24 h incubation, while in MDA-MB-231 cells, a significant decrease was noticed after 48 and 72 h only (Figure 3A,B, followed by a densitometry analysis relative to loading control, GAPDH—Figure 3C,D, respectively). Thus, a multi-faceted mechanism of action of the TMPyP4 was revealed both at the level of activity (probably due to postulated stabilization of G4 and telomerase access prevention) and at the hTERT expression down-regulation. Importantly, the fact concerning the ability of TMPyP4 to diminish both telomerase activity and expression was previously suggested [15]. Interestingly, BIBR 1532 and DOX did not show any significant effect in hTERT protein accumulation in MCF7 cells while in MDA-MB-231 cells the catalytic telomerase subunit was also repressed after treatment with BIBR 1532 (48 and 72 h) as well as DOX (48 h) (Figure 3A–D).

### 2.3. TMPyP4—Mechanism of Action

Further studies of the mechanism of action of TMPyP4 showed some genotoxic (or cytostatic) effect on studied cells in a clonogenic assay (Figure 4). Low concentrations of TMPyP4 (i.e., 0.5 and 5 µM) did not provoke significant effects in MCF7 cells, but when the compound was used in a concentration of 10 or 20 µM a significant decrease (20 or 50%, respectively) in cell viability was observed (Figure 4). Clonogenic assay seemed to be more sensitive than MTT although the cells were incubated with the compound for the same time, i.e., 72 h, followed by another 10 days of growth monitoring. When MDA-MB-231 cells were analyzed, a similar, non-significant effect was observed after 0.5, 5, or 10 µM TMPyP4 was used. However, in higher concentrations, the porphyrin drastically abolished the survival of the cells (Figure 4).

Interestingly, flow cytometry analysis revealed that none of the applied concentrations of the porphyrin provoked cell apoptosis (Figure 5). However, MCF7 cells treated with TMPyP4 showed some non-significant and dose-dependent increase in the accumulation of G0/G1 cells (Figure 5A1). Additionally, a significant decrease in the S phase after treatment with 50 μM porphyrin was observed. (We did not observe any significant cell-cycle alterations when cells were treated for 24 or 48 h; see Appendix A). At the same time, MDA-MB-231 cells did not show any alterations when treated with 10 μM TMPyP4, but when treated with 20 or 50 μM porphyrin they revealed a significant accumulation of cells in S and G2/M that was accompanied by a significant decrease of the number of cells in G0/G1 phase. These effects were observed after 72 h incubation time (Figure 5A2). Interestingly, we did not see any significant proapoptotic effects in MCF7 cells when treated with a combination of TMPyP4 and DOX, but the population of S phase cells was significantly reduced (accompanied by G0/G1 increase) in all applied concentrations (Figure 5A1). In the case of MDA-MB-231 cells, the influence of both compounds was manifested by an intermediate effect (relative to samples treated with the compounds alone) suggesting no additive effect (Figure 5A2).

Furthermore, we performed an analysis of the level of proteins associated with cell-cycle phases. Cyclin D1 and Cyclin E regulate the transition from G1 to S phase whereas the progression from G2 to M phase is regulated by Cyclin B1. In our study, we did not see any significant changes in B1, D1, and E cyclins when cells were treated with 10 μM TMPyP4 in both cell lines (Figure 5C1–E1,C2–E2 for MCF7 and MDA-MB-231, respectively). However, when MCF7 cells were exposed to higher concentration of the test substance (20 μM), a significant decrease in cyclin D1 and E accumulation was observed (Figure 5C1–E1) which may reflect cell-cycle arrest in G1 that was observed in flow cytometry but as non-significant (Figure 5A1). The observed induced accumulation of MCF7 cells in G0/G1 phase was accompanied by a dose-dependent decrease of Cyclin D and Cyclin E levels. Interestingly, the 50 μM porphyrin provoked a significant decrease in all the cyclin accumulation, but this effect might be non-specific since we observed that high concentration of TMPyP4 (50 μM) led to a degradation of GAPDH (Figure 5B1). In MDA-MB-231 cells, the porphyrin effect was dose-dependent, and all the cyclins were down-regulated (Figure 5B2) and at 50 μM TMPyP4 the cyclin B1 was almost totally abolished. The significant accumulation of MDA-MB-231 cells in S and G2/M was accompanied by a significant decrease of Cyclin B1 (Figure 5B2,C2).

### 2.4. Effect of TMPyP4 on DNA Damage and Repair Signaling

Further detailed studies were supposed to reveal the mechanism of the potential association of telomerase down-regulation and its contribution to the induction of DNA damage/repair pathway. Thus, we continued with immunoidentification of proteins contributing to DNA damage and repair signaling. For this reason, we checked a dose-dependent effect of TMPyP4 (concentration range 0.5, 5, 10, or 20 μM) in both studies cell lines after 72 h treatment. Samples treated with 0.1 or 0.2 μM DOX (double strand break/DSB inducer) as well as 5 or 10 μM cisplatin (another DSB inducer) were used as positive controls to verify the specificity of the effect. Similarly, cells incubated with 2 μM DOX for 2 h were used as a reference positive sample in experiments where no significant effect or very low level of target proteins were expected.

Both cell lines were assessed, and in MCF7 we observed that the porphyrin was efficient in hTERT down-regulation. However, it did not provoke a response in histone γ-H2A.X (phosphorylated at Ser139) accumulation (Figure 6A) that is perceived as a molecular marker of DNA damage (double strand breaks) [19]. Similarly, it did not provoke phosphorylation of Serine 15 in p53, which is the primary target of the DNA damage response. In MDA-MB-231 cells, hTERT down-regulation was also efficient and dose-dependent (Figure 6B). Interestingly, the basal level of γ-H2A.X was higher, and, surprisingly, porphyrin treatment contributed to the diminished accumulation of this protein. A similar effect was triggered in p53 Ser15 phosphorylation, while the signal in positive control samples was much higher than in control samples. In addition, again, MCF7 cells showed much lower basal level of phosphorylated p53 (Ser15) while in MDA-MB-231 cells TMPyP4 provoked down-regulation of Ser15 phosphorylation (Figure 6B).

Similarly, we could not see any induction of DNA repair-associated pathways, i.e., ATM (key mediator of the histone γ-H2A.X phosphorylation at Ser139) or DNA-PKC phosphorylation, which suggested no damage in the DNA of cells exposed to TMPyP4. Additionally, porphyrin treatment did not provoke activation of PARP (except for positive controls) in none of the studied cells. We could not see any changes in the p21 signal while in both cell lines, positive controls demonstrated increased accumulation of this senescence marker and cell-cycle inhibitor. The proliferation marker, PCNA, was unaltered in MDA-MB-231 cells, but it was diminished in MCF7 cells.

### 2.5. TMPyP4 Affects Adhesion and Migration Processes

During all experiments, we observed some issues with the adhesion of the studied cells. Interestingly, similar observations were revealed during previous studies on the biological activity of TMPyP4 in a broad range of concentrations [20]. To evaluate the effect, we performed some functional tests, including scratch assay and adhesion test. To avoid cytotoxic or cytostatic effects, we used low concentrations of TMPyP4, i.e., 0.5, and 5µM. In both cases, we observed significant inhibition of wound-growing by both cell lines (Figure 7). Just for clarification, none of the concentrations provoked any significant cell survival alteration (MTT assay, data not shown). To verify the adhesion properties of the studied cells, we performed a functional adhesion test. In this assay, cells showed lowered adhesion potential that was dose-dependent in the range of concentrations 10, 20, and 50 µM. The effect observed in MDA-MB-231 cells was clearer; however, in both cell lines, the morphology was significantly altered, cells were round-shaped and revealed some difficulties witch the attachment (Figure 8). For this reason, we continued our studies towards the analysis of the mechanism of action of TMPyP4. It is known that one of the most critical pathways in cell communication, adhesion, and migration is the one associated with integrin beta-1 signaling. Interestingly, we observed a significant decrease of integrin beta-1, FAK (Focal Adhesion Kinase), and paxillin as well as phosphorylation status of the two later proteins in MCF7 cells (Figure 9A1,B1). Interestingly, the effect observed in MDA-MB-231 cells was also significant but only at 72 h incubation time (Figure 9A2,B2)—which corresponds to the hTERT down-regulation efficacy by TMPyP4 (Figure 3).

To verify the specificity of action of studied compounds and potential proapoptotic effect, we assessed caspase-3 activation, and we did not reveal the activation of apoptosis (Figure 9). It is worth noting that due to a mutation in caspase-3 coding gene, we could not detect the protein using the antibody in MCF7 cells.

To summarize, there was a very limited effect on cell survival in the MTT test, and there was no significant effect on the caspase activation in MDA-MB-231 cells. Altogether these results indicated no contribution of TMPyP4 to induction of cancer cell death or senescence in a short incubation time (up to 72 h) and a wide range of concentration (0.5 to 50 μM). Additionally, no accumulation effect of TMPyP4 treatment with DOX was shown—we could only see the rather dominant effect of DOX.

## 3. Discussion

### 3.1. TMPyP4 as a Tool in Targeting Telomerase in Breast-Cancer Therapy

Due to increasing cancer morbidity, new therapy strategies are being developed. Concerning the fact that there are no perfect qualitative markers that would serve specifically enough as a molecular target, the best way to conquer the disease is to apply an adjuvant therapy. It seems that the simultaneous targeting of various mechanisms might contribute to sensitization and/or elimination of cancer cells. One of the most exclusive and consequently attractive candidates among molecular cancer targets is telomerase. This enzyme provides cancer cell immortality, but it may be efficiently stopped using different methods [21]. It is associated with the fact that the contribution of the enzymatic complex to cell survival goes beyond the ability to prevent telomere shrinking. Numerous reports are showing some other, non-canonical functions of telomerase [2,22].

The simplest way to reveal the function of telomerase is to turn it off and verify the effects. Numerous studies showed that down-regulation of the whole complex or individual subunits brings some critical alteration in cell metabolism [23]. TMPyP4 is one of the compounds that shows some potential in telomerase-targeting and cancer elimination. The effect of the compound is mediated through its ability to bind DNA and stabilize the guanine-quadruplexes. Consequently, it can interfere with the ability of telomerase to extend telomeres [7]. Other mechanisms of telomerase-targeting are based on down-regulation of the key subunits, i.e., hTERT or hTR [23], but immunotherapy, gene therapy, and other approaches are also being developed [24].

The experiments were performed with the use of telomerase-positive breast-cancer cells MCF7 and MDA-MB-231. These two breast-cancer models have very different molecular characteristics, with MCF7 being ER+ while MDA is a triple-negative breast-cancer model. In this study, we show that TMPyP4 revealed a cytostatic effect on breast-cancer MCF7 and MDA-MB-231 cells, despite their relatively high telomerase activity (Figure 1). Interestingly, in a non-cancer cell line MCF-12A that shows much lower expression of hTERT, a significantly weaker cytostatic effect was observed. We showed that micromolar concentrations of the porphyrin caused a significant decrease in both hTERT expression (mRNA) (Figure 2A,B) as well as telomerase activity (Figure 2C,D) in both cancer cell lines. It is worth noting that the basal hTERT level in MCF7 cells was much higher than in MDA-MB-231 (Figure 2E) which might explain diverse effects. Importantly, TMPyP4 also significantly down-regulated accumulation of hTERT protein in both cancer cell lines (Figure 3). This effect cannot be explained only by direct G4 stabilization, but suggests the existence of different molecular mechanisms triggered by TMPyP4 biological activity.

Interestingly, previous studies have shown that TMPyP4 decreased c-MYC expression at the RNA and protein levels [15]. This may result from the fact that both FAK and Myc are closely located at 8q24 locus [25]. Inhibition of the integrin/FAK signaling axis and c-Myc synergistically disrupts ovarian cancer malignancy. It is worth noting that the c-MYC promoter region also contains guanine-rich DNA that can form G4 structures that makes it a good target for TMPyP4 [26]. Since this proto-oncogene plays a significant role in the expression of hTERT, it might explain the reduction of telomerase activity by TMPyP4. In our studies, we used TMPyP4 also in combination with a known chemotherapeutic agent—doxorubicin. Its effectiveness is limited by serious side effects caused by high concentrations of this drug [27]. We hypothesized that combination therapy could be more efficient (even at low DOX dose) since telomerase-targeting might appear a good tactic in cancer cell sensitization. It was reported by Shi et al. [28] that a combination of telomerase inhibitor BIBR1532 and paclitaxel acted synergistically and induced growth inhibition in breast-cancer cell lines. Surprisingly, in our studies, simultaneous treatment of breast-cancer cells with TMPyP4 and doxorubicin did not show any cumulative cytostatic or cytotoxic effect (Figure 1; flow cytometry results were more informative and will be discussed below). However, the time exposure was a maximum of 72 h, which might be a limiting factor. Interestingly, some studies show that inhibition of telomerase itself does not always result in the induction of cell death, but affects the number of mechanisms directly and indirectly associated with the presence of telomeres and telomerase, including mitochondria protection [29], proliferation control [30], or even DNA repair support [31]. To find the pathways regulated by TMPyP4, we did perform further analysis of the effects of the porphyrin on breast-cancer cells. Since the toxic properties measured with MTT assay seemed to be limited, we assessed the effect of TMPyP4 on colony-forming properties of studied cell lines (Figure 4). The results suggested a genotoxic effect, but since no alteration in γ-H2A.X was shown, it seems again to trigger some other effects, i.e., cytostatic. Additionally, the difference between MTT and clonogenic experiments might also result from the fact that both experiments assess slightly different mechanisms. It is worth noting that the clonogenic assay seems to be more sensitive due to a lower starting number of cells compared to MTT (200 vs. 5000 cells, respectively) [32]. Interestingly, TMPyP4 was already shown to reveal a significantly different result, relative to the studied cell lines (relative to basal telomerase activity) [33] suggesting that cell death caused by TMPyP4 might be mediated by both telomerase activity loss or sufficient telomere shortening. It is worth noting that TMPyP4 concentrations used in those experiments and most studies are in the range of 50–100 µM. As we showed, 50 µM is a concentration that caused a significant cytostatic effect in MTT assay in all studied cell lines.

### 3.2. Contribution of TMPyP4 to Cell-Cycle Modulation

Interestingly, flow cytometry assay revealed only slight alteration in the MCF7 cell cycle (Figure 5A) while in MDA-MB-231 this effect is more evident and significant at the concentrations of 20 and 50 µM but also without induction of apoptosis (Figure 5B). Surprisingly, when a combination of the two compounds, i.e., TMPyP4 and DOX, was applied, not only was there no sensitization of the cells to DOX, but significant attenuation of the proapoptotic action of DOX was observed (Figure 5A,B, respectively). Additionally, MCF7 co-treatment seems to result in the accumulation of the cells in G0/G1 phase. The response of MDA-MB-231 cells was more indefinite—subpopulation content was altered with some significant reduction of the apoptotic fraction. Both cell lines showed slightly different results in flow cytometry assessment, and MDA-MB-231 cells seemed to be more susceptible to cell-cycle alteration than MCF7, which could result from a lower basal telomerase level (Figure 2E). Altogether, these observations confirm no apoptosis after TMPyP4 treatment in the studied breast-cancer cells. Thus, we decided to assess the cell-cycle-associated proteins. However, no significant changes in B1, D1, or E cyclins were revealed when cells were treated with 10 μM TMPyP4 in both cell lines (Figure 5C,D for MCF7 and MDA-MB-231, respectively). However, when MCF7 cells were exposed to higher concentrations of the test substance, a significant decrease in D1 accumulation was observed, which may reflect cell-cycle arrest in G1 that was observed in flow cytometry (Figure 5A). Similarly, dose-dependent effect on B1 and E cyclins suggested a contribution to the regulation of the cell cycle. Interestingly, the 50 μM porphyrin provoked a significant decrease in all the cyclin accumulation, but this effect might be non-specific, especially since, as mentioned above, we also observed the degradation of the loading control, i.e., GAPDH, and that could be associated with an energetic crisis leading to cell death [34].

Since apoptosis was expected, caspase-3 activation was assessed in MDA-MB-231, but no alteration was reported. In MCF7 (caspase-3 negative) it was impossible to assess caspase-3, due to issues mentioned in the Results section. For this reason, cell-cycle assessment was performed at 24 and 48 h, revealing no significant alterations, including no apoptosis (see Appendix A). Interestingly, in MDA-MB-231 cells, the cell-cycle alterations were more evident but again only at concentrations 20 and 50 µM. A clear decrease of the G0/G1 phase was observed that was accompanied by an increase in S and G2/M phase but without apoptosis induction. Interestingly, apoptosis provoked by DOX was again attenuated by a combination with TMPyP4. Those differences may result from the fact that DOX arrests MCF7 cells at G1/S and G2/M checkpoints, whereas MDA-MB-231 cells are arrested at G2/M only [35]. Consequently, TMPyP4 by inducing G0/G1 phase accumulation might attenuate the effect of DOX. Those cell-cycle alterations were expressed by a dose-dependent decrease in the accumulation of cyclins B1, D1, and E.

### 3.3. TMPyP4 in DNA Damage/Repair Pathway

Consequently, we decided to analyze the DNA damage/repair pathways (Figure 6). We demonstrated that down-regulation of hTERT in the studied cell lines did not provoke either γ-H2A.X or p-Ser15 p53 phosphorylation. Similarly, no phosphorylation of DNA-PKCs or ATM was observed, suggesting no contribution to DNA damage/repair signaling. However, the possible association may not be excluded. As demonstrated by Tauchi et al. [36], another G-4 inhibitor, telomestatin, could activate the ATM and Chk2, suggesting that telomere dysfunction induced by telomestatin activates the ATM-dependent DNA damage response in human leukemia cells. Similarly, there was no activation of PARP, and none of the studied proliferation markers, i.e., PCNA, show alterations (Figure 6). However, if the effect were based on telomerase attrition only, multiple cell cycles would be required for the telomerase inhibition-mediated end-replication problem to induce sufficient telomere shortening and trigger a DNA damage response at the chromosome ends. Interestingly, the studies of another potential telomerase inhibitor, telomestatin, showed no lag time between treatment and induction of DNA damage [37]. Additionally, we observed no effect on the cell-cycle inhibitor, i.e., p21 (direct transcriptional target of p53) accumulation which suggested no pro-aging activity in the short-term incubation, but it is known that telomerase inhibition may induce cell-cycle arrest through p21 activation [37].

Both studied cell lines are different, as described in the text. It is worth noting that concerning the DNA damage response (DDR) assessment, MCF7 revealed a very low basal level of phosphorylated Ser15 in p53 as well as γ-H2A.X while both proteins were highly elevated in control MDA-MB-231 cells. However, no DDR was induced in none of the cell lines after TMPyP4 treatment. Concerning the cell-cycle assessment, the effects were different in both cell lines in a semi-quantitative WB analysis, but the trend was similar. Altogether, it might be due to different resistance/sensitivity level to the compound resulting from different signaling in both cell lines. Altogether, MDA-MB-231 cells were more resistant in MTT and, similarly, the cell cycle of those cells was only moderately affected when TMPyP4 was involved.

### 3.4. Contribution of TMPyP4 to Functional Impairment of Migration and Adhesion

Most importantly, TMPyP4 treatment provoked significant inhibition of migration and adhesion properties of cancer cells (Figure 7). Additionally, the porphyrin significantly inhibited the adhesion potential of studied cancer cell lines in a dose- and time-dependent manner (Figure 8). It is not the first time that such an activity is postulated. However, Zheng et al. [20] declared a dose-dependent effect in a very low range of concentrations (0.125–2 µM) in A549 and U2OS cells. For this reason, we decided to assess the potential pathway engaged in this mechanism.

Consequently, we revealed that in MCF7 cell line, TMPyP4 significantly diminished integrin beta-1-mediated pathway (through the decrease of both protein level and phosphorylation of FAK and paxillin) while another telomerase inhibitor, BIBR1532, did not alter this pathway so substantially (Figure 9A1,B1). It may be that the pleiotropic activity of the porphyrin is revealed and thus the effect is more rapid and more significant, and may not be mediated through hTERT and telomerase modulation. At the same time, the significantly weaker effect was observed in MDA-MB-231, which also demonstrated a less efficient response when assessing telomerase activity and hTERT expression (at both mRNA and protein level) (Figure 2B,D). Liu et al. [13] postulated the relationship between telomerase and cell adhesion and migration capacities of cancer cells. They showed that hTERT contributed to the promotion of cell adhesion and migration, not due to its enzymatic activity, but as the result of its non-canonical functions. Moreover, sequencing of the genome of cells overexpressing hTERT showed a positive correlation between the expression of hTERT and the expression of genes involved in cell adhesion and the organization of the extracellular matrix [13]. As with our observations, analysis of the impact of TMPyP4 on the migration of melanoma cells (B78-H1) showed inhibition of cell migration during 24 h treatment [37]. The ability of TMPyP4 to attenuate the cell adhesion and migration properties can contribute to a reduction in the metastatic potential of cancer cells. Importantly, the integrin β-1/FAK pathway is engaged in cell proliferation [38], and it may be that the inhibition of this pathway is associated with the cytostatic effect of TMPyP4 in breast-cancer cells. However, previously mentioned studies of Zheng et al. [20] revealed that low concentrations of TMPyP4 (<0.5 μM) could increase adhesion and migration of cancer cells. Thus, we should be very careful when planning an anticancer strategy approach based on telomerase inhibitors.

## 4. Materials and Methods

### 4.1. Cell Lines and Study Compounds

These two studied breast-cancer models have very different molecular characteristics, including ER status (ER+, MCF7, and triple-negative MDA-MB-231). Importantly, they also reveal a different metastatic potential (MCF7, low and MDA-MB-231, high, respectively). Additionally, a non-tumorigenic epithelial cell line, MCF-12A, was also used as a reference cell line that is reported to show residual telomerase expression and activity. Both human breast adenocarcinoma cell lines MCF7 (ER+, p53 wild type) and MDA-MB-231 (tripel-negative, p53 mutant) and non-tumorigenic epithelial cell line MCF12A were obtained from the American Type Culture Collection (HTB-22, HTB-26, CRL-10782).

The MCF7 and MDA-MB-231 cells were cultured in RPMI-1640 medium (Biochrom GmbH/Merck Millipore, Berlin, Germany) supplemented with 10% fetal bovine serum (Biochrom GmbH/Merck Millipore, Berlin, Germany) in an atmosphere of 5% CO_2_ and 100% humidity at 37 °C. The MCF12A cells were cultured in Ham’s F-12/DMEM medium supplemented with 20 ng/mL human epidermal growth factor, 100 ng/mL cholera toxin, 0.01 mg/mL bovine insulin, 500 ng/mL hydrocortisone and 5% horse serum. The cells were passaged with medium changes every 3–4 days.

TMPyP4, 5,10,15,20-Tetrakis(1-methylpyridinium-4-yl)porphyrin tetra(p-toluenesulfonate), was purchased from AbcamBiochemicals (Cambridge, UK). Doxorubicin hydrochloride and cisplatin were obtained from Sigma–Aldrich (St. Louis, MO, USA), and BIBR 1532 from Cayman Chemical (Ann Arbor, MI, USA). BIBR 1532 is a mixed-type non-competitive inhibitor of telomerase (IC_50_ = 93 nM) that specifically targets the telomerase reverse transcriptase catalytic subunit, hTERT that is followed by induction of senescence or apoptosis in cancer cells [39]. All procedures and storage were performed with minimal exposure to light due to TMPyP4 susceptibility to light.

### 4.2. Cytotoxicity Assay

Cytotoxicity of TMPyP4, doxorubicin alone, or both compounds in combination, was assessed using the MTT Proliferation Assay as previously described [40]. Briefly, a total of 5 × 10^3^ MCF7, MDA-MB-231 or MCF-12A cells were seeded into each well of 96-well plates in a total medium volume of 100 μL/well. Cells were exposed for 24, 48, and 72 h to doxorubicin (0.1 μM) or 10–100 μM of TMPyP4 alone or in combination with 0.1 μM of doxorubicin. The solvent, H_2_O, was also applied as a control. Subsequently, 10 μL of MTT solution (5 mg/mL) (Sigma–Aldrich, St. Louis, MO, USA) was added to each well. The cells were incubated at 37 °C for 4 h followed by 100 μL of solubilization buffer (10% SDS in 0.01 M HCl) addition. Cell viability was quantified using a LabsystemsMultiscan RC spectrophotometer. Three separate experiments were performed, with eight repeats for each concentration. IC_50_ values were calculated using CalcuSyn (Biosoft, Cambridge, UK), and the standard deviation was calculated using Excel software (Microsoft, WA, USA).

### 4.3. Quantitative Assessment of hTERT Expression

mRNA expression of hTERT was investigated in MCF7 and MDA-MB-231 cells. 5 × 10^4^ cells were seeded into 60 mm plates and exposed for 72 h to 10, 20 and 50 μM of TMPyP4 alone or in combination with 0.1 μM of doxorubicin. RNA isolation was done using the TRI Reagent Protocol (Sigma, St. Louis, MO, USA) according to the manufacturer’s instructions. Concentration and quality of extracted RNA were evaluated by optical density measurement (A260/A280 ratio) with Biophotometer Plus (Eppendorf, Hamburg, Germany). 500 ng of total RNA was reverse transcribed using oligo(dT) primers and Transcriptor First Strand cDNA Synthesis Kit (Roche Diagnostics, IN, USA). Real-time PCR was carried out using SYBR Green-based reaction mixes (LightCycler^®^ FastStart DNA Master SYBR Green I, Roche Diagnostics, IN, USA). Primers sequences and annealing temperatures used: hTERT: F: GCCGATTGTGAACATGGACT, R: CACCCTCGAGGTGAGACG (60 °C); GAPDH: F: TTCGTCATGGGTGTGAACC, R: GATGATGTTCTGGAGAGCCC (60 °C) and a LightCycler^®^ 2.0 (Roche Diagnostics, IN, USA) was used as follows: 95 °C, 10 min; (94 °C, 15 s; Ta, 15 s; 72 °C, 15 s) ×40; 72 °C, 5 min. The GAPDH expression was provided as an internal reference gene (housekeeping gene) to normalize the expression of the hTERT.

### 4.4. Telomerase Activity

The effect of TMPyP4 on telomerase activity in breast-cancer cells was assessed using the quantitative TRAPEZE^®^ RT Telomerase Detection Kit (Merck Millipore, Darmstadt, Germany), a modified original telomeric repeat amplification protocol (TRAP). All procedures were performed according to the manufacturer’s protocol, as previously described [40]. Briefly, a cell extract was prepared from MCF7 and MDA-MB-231 cells treated with 10, 20 or 50 µM TMPyP4 alone or in combination with 0.1 μM doxorubicin for 72 h. For each assay, 1 µg of protein cell extract was used. After 30 min incubation at 30 °C for primer extension, the PCR cycling conditions were: 95 °C for 2 min followed by 45 cycles at 94 °C for 15 s, 59 °C for 60 s and 45 °C for 10 s. The results were quantitated using fluorescein-labeled Amplifluor^®^RP primers. A dilution series of TSR8 control template was prepared in CHAPS lysis buffer to serve as a standard curve. Reactions were set up in triplicate. The fluorescence emission produced is directly proportional to the amount of TRAP products generated. Heat-inactivated cell extracts and lysis buffer were tested as a negative control. The quantity (amoles) of extended telomerase substrate produced in each well from the telomerase activity of cell was determined from a linear plot of the log_10_ of the quantities (amoles) of TSR8 control template standards versus the Ct values for their wells. The mean value of these quantities for the three replicate wells for each sample was calculated.

### 4.5. Western Blot Analysis

Cells were treated with TMPyP4, doxorubicin, BIBR 1532, or cisplatin as indicated. Whole-cell extracts were prepared using RIPA buffer (50 mMTris–HCl, pH 8.0, 150 mMNaCl, 1% NP40, 0.1% SDS, 100 mM PMSF, 25 μg/mL Na3VO4, 25 μg/mL NaF, 25 μg/mL leupeptin and 25 μg/mL aprotinin). The protein concentration was measured using the Bradford assay (Sigma–Aldrich, St. Louis, MO, USA) and 40 μg of each extract was loaded onto SDS–PAGE gels. Western blotting was performed by a standard procedure using a PVDF membrane (Pierce Biotechnology, Rockford, USA). The transfer was followed by blocking the membrane with 5% skimmed milk or BSA in PBS-T. The following primary antibodies were used for detection: anti-hTERT (Novus Biologicals), anti-GAPDH, anti-Cyclin B1, anti-Cyclin D1, anti-Cyclin E (Santa Cruz Biotechnology); anti- γ-H2A.X (Ser139), anti-p-Ser15 p53, anti-pATM S1981, anti-DNA-PKCs, anti-PARP, anti-PCNA, anti-p21, anti-tubulin, anti-β1integrin, anti-FAK, anti-p-FAK Y397, anti-p-FAK Y576/577, anti-paxillin, anti-p-paxillin, anti-caspase 3 (Cell Signaling Technology); 1 μg/mL of each primary antibody was used in the blotting solution. After removal of the antibodies, anti-rabbit IgG and anti-mouse IgG (Cell Signaling Technology) secondary antibodies labeled with horseradish peroxidase, were added. The proteins were visualized using SuperSignal^®^ West Pico Chemiluminescent Substrate and CL-X Posure™ film (Pierce Biotechnology, Rockford, USA) or chemiluminescence camera using VisionWorks software (UVP, Upland, CA, USA).

### 4.6. Colony Formation Assay

The clonogenic assay was used to determine the effectiveness of cytotoxic agents by assessing colony formation [41]. MCF7 and MDA-MB-231 cells were harvested, counted, and seeded into 60 mm plates in the concentration of 200 cells on each plate. After 24 h of incubation in a CO_2_ incubator at 37 °C cells were exposed to 0.5–20 μM of TMPyP4. After 72 h the medium was removed, and cells were washed twice with PBS. Then fresh medium without a cytotoxic agent was added, and cells were maintained for 10 days followed by a 10 min fixation in methanol and Giemsa staining (Sigma–Aldrich, St. Louis, MO, USA) for 1 h. Stained colonies were counted. Three separate experiments were performed for each cell line.

### 4.7. Cell-Cycle Analysis

To analyze the influence of TMPyP4 alone and in combination with doxorubicin on the cell cycle, a flow cytometry analysis using propidium iodide was performed. The addition of propidium iodide allows subsequent stages of apoptosis and eventual cell death to be distinguished [42]. Consequently, in the presented study, cells with reduced PI staining of DNA (sub-G1 fraction) were considered apoptotic. Even if not very precise, this method enabled quick verification of the apoptotic population. After 72 h exposure to the drugs cells were collected using 0.25% trypsin (Sigma, St. Louis, MO, USA) and cell pellets were washed twice in PBS, centrifuged at 300 g and re-suspended in 200 μL of reaction mixture containing: 50 μg/mL propidium iodide and 10 mg/mL of RNase A (Sigma, St. Louis, MO, USA) in PBS. After 1 h of incubation in the dark at 37 °C, the samples were analyzed by flow cytometry (Becton–Dickinson, FACScan).

### 4.8. Scratch Assay

In-vitro wound healing assay for studying cell migration was performed using MCF7 and MDA-MB-231 cells in cultures that contained TMPyP4. Briefly, 2 × 10^5^ cells/well were seeded on 6-well plates for both cell lines and allowed a 70% confluence. A 1-mm width linear wound was created across the center of each well with a plastic tip. Wounded monolayers were then washed three times with medium to remove cell debris and incubated in the presence of 0.5 and 5μM TMPyP4. Photographs of the scratches were captured on day 0 and again 24, 48 and 72 h later under phase contrast microscope (Carl Zeiss, Oberkochen, Germany). The micrographs of the scratch wound healing assay are representative of three independent experiments.

### 4.9. Cell Adhesion Assessment

MCF7 and MDA-MB-231 cells were grown to 70% confluence in 60 mm dishes. After 24-h incubation cells were exposed to 10, 20 or 50 μM TMPyP4 for 24 h. Then cells were harvested, washed twice with warm PBS and counted. 5 × 10^5^ cells were seeded in 60 mm dishes with the warm medium. Cells were analyzed under the phase contrast microscope and photographed after 15, 30, 60 min, 3 h, and 18 h.

### 4.10. Statistics

Statistical significance of differences between the groups (triplicates) was evaluated by two-tailed, unpaired *t*-test.

## 5. Conclusions

Telomerase inhibitors display properties towards the selective killing of cancer cells in a variety of biological models by modulating different molecular pathways. However, some of those compounds may reveal additional, unexpected side effects. These may be caused by the fact that telomerase is known for its non-canonical activities, and targeting this enzyme may also provoke some surprising effects. Secondly, potential telomerase inhibitors could trigger some consequences that may or may not be mediated by telomerase down-regulation due to their unknown biological potential. It may be that TMPyP4 belongs to a group of such compounds. In general, TMPyP4 is mainly recognized for its ability to target telomeres and thus inhibit telomerase and eliminate cancer cells; however, it appears that its biological potential is much wider.

In the present study, we could not identify a direct correlation between telomerase inhibition with TMPyP4 and cell-cycle modulation. It may be that verification of this association might require a longer duration of exposure to reveal telomere-length-dependent effects.

Conversely, it may be that TMPyP4 provokes specific alterations within the cancer cell mechanism in a telomerase-independent way. For this reason, short time intervals of the exposition may be beneficial to reveal pathways that are engaged in cell migration and adhesion or gene expression control. Nevertheless, in the context of adjuvant cancer therapy, it may be that it is not primarily about targeting telomerase or telomeres, but about sensitization of cancer cells to drugs, thus weakening their migration and adhesion properties that would limit their aggressive potential.

## Figures and Tables

**Figure 1 ijms-20-02670-f001:**
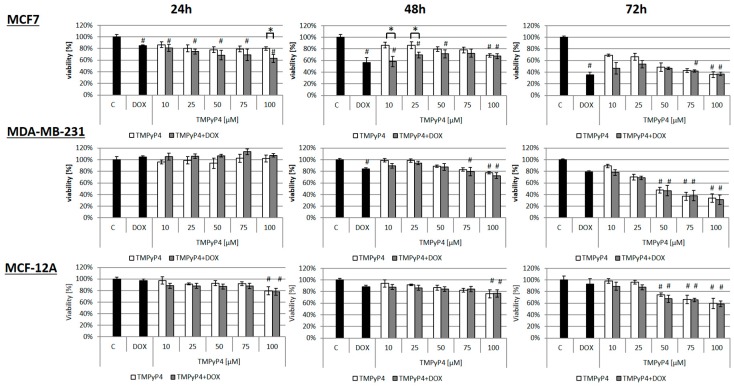
Cancer cell survival after exposure to telomerase inhibitor, TMPyP4. MCF7, MDA-MB-231, and MCF-12A cells were exposed to different concentrations of TMPyP4 (10–100 µM; open bars), doxorubicin (DOX, 0.1 µM) or a combination of both compounds (grey bars) for 24, 48, or 72 h. The viability was assessed using MTT assay. Cell survival was assessed relative to the control sample (C). * *p* < 0.05, TMPyP4 relative to TMPyP4+DOX; # *p* < 0.05, relative to control sample. Tests were performed in biological triplicates (each replicate consisted of 8 technical replicates/wells).

**Figure 2 ijms-20-02670-f002:**
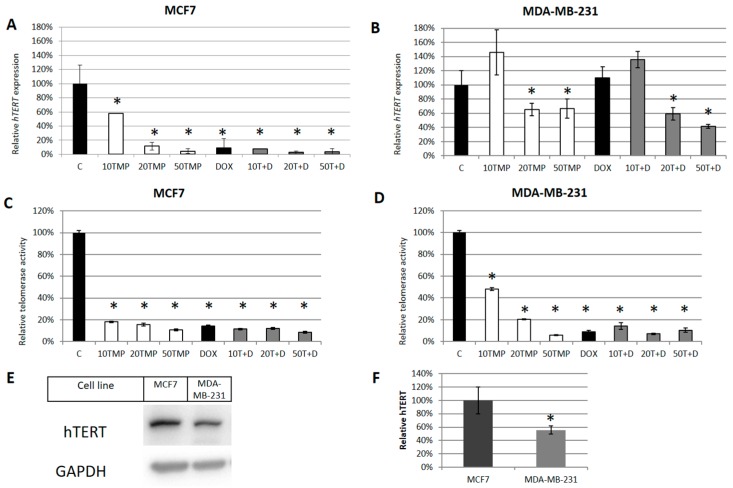
TMPyP4 alters telomerase expression and activity. The contribution of TMPyP4 to telomerase expression in MCF7 (**A**) and MDA-MB-231 cells (**B**) was assessed using qPCR. GAPDH (Glyceraldehyde-3-Phosphate Dehydrogenase) was used as a housekeeping/relative gene. Cells were treated for 72 h with TMPyP4 (10, 20 or 50 μM), DOX (0.1 μM) or a combination of those two compounds, total RNA was isolated, poly(A+)mRNA was reversely transcribed, and hTERT gene expression was assessed. Similarly, the influence of studied compounds on telomerase activity was analyzed using telomeric repeat amplification protocol (TRAP) assay in MCF7 (**C**) and MDA-MB-231 cells (**D**); cell treatment, as mentioned in the expression study. The basal level of hTERT (**E**) in studied cell lines was assessed using western blot after 24 h from seeding and densitometry analysis, relative to loading control GAPDH, was performed to reveal the difference (**F**). The same amount of total protein from both cell lines was used, i.e., 40 µg. * *p* < 0.05, relative to control sample.

**Figure 3 ijms-20-02670-f003:**
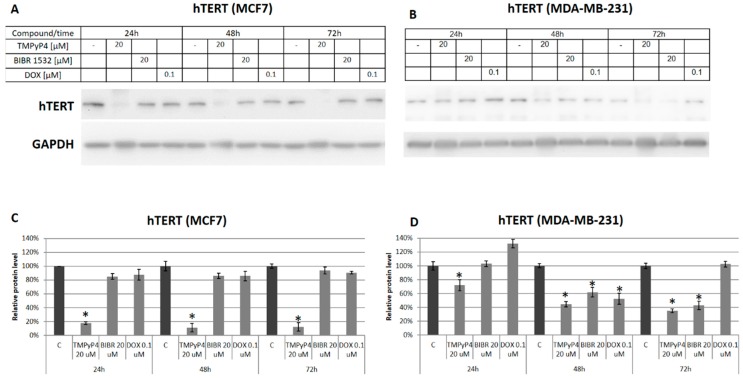
TMPyP4 efficiently decreases hTERT accumulation in a time-dependent manner. MCF7 (**A**) and MDA-MB-231 (**B**) cells were subjected to treatment with TMPyP4 (20 µM), BIBR 1532 (20 µM) or DOX (0.1 µM) in a time-course experiment. After individual time intervals (24, 48, or 72 h) cells were lysed, and immunoidentification of hTERT was performed. The concentration of compounds was adjusted according to MTT assay. GAPDH was used as a loading control. Densitometry analysis of the western blot results was performed relative to the loading control, GAPDH (**C**,**D**). * *p* < 0.05, relative to control sample.

**Figure 4 ijms-20-02670-f004:**
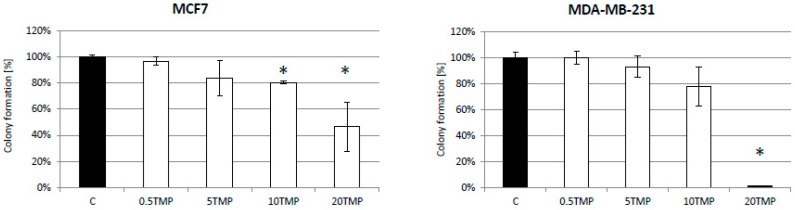
TMPyP4 reveals a genotoxic effect. Colony formation assessment was performed after 72 h treatment with TMPyP4 (0.5, 5, 10 or 20 µM) followed by 10 days of culture. The cells were seeded on 6-well plates at 200 cells per well. The experiment was performed in triplicates. * *p* < 0.05, relative to control sample.

**Figure 5 ijms-20-02670-f005:**
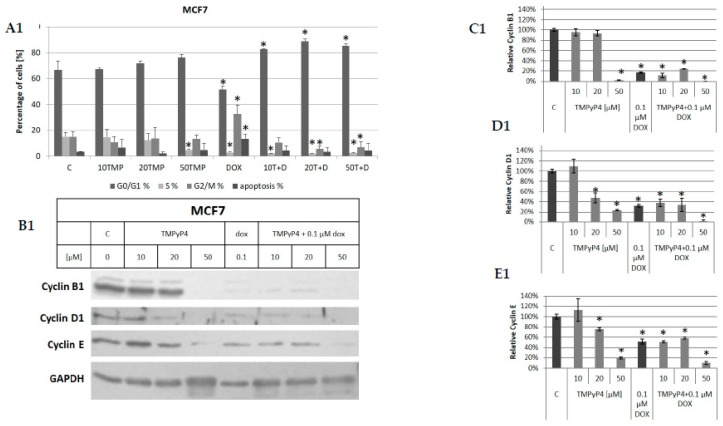
The contribution of TMPyP4 to cell-cycle modification. MCF7 (5.1) and MDA-MB-231 (5.2) were subjected to flow cytometry assessment (**A**), followed by immunodetection of cell cycle associated proteins (**B**) and densitometry analysis of the blots i.e. Cyclin B1 (**C**), Cyclin D1 (**D**), Cyclin E (**E**), respectively. Both studied cell lines were subjected to the analysis of cell cycle after treatment with 10, 20, 50 μM TMPyP4, 0.1 μM DOX or a combination of both compounds for 72 h since no significant effects were revealed in shorter time interval neither in MTT nor in a clonogenic assay. Detection of cell-cycle phase was performed in MCF7 (5.1) and MDA-MB-231 cells (5.2) using propidium iodide staining and flow cytometry. Immunodetection of cyclins B1, D1, and E after 72 h treatment with TMPyP4 (0, 10, 20 or 50 μM), DOX (0.1 μM) or a combination of these compounds was performed using western blot in MCF7 (5**B1**) as well as MDA-MB-231 cells (5**B2**) followed by densitometry analysis (5**C1**,**D1**,**E1**,5**C2**, **D2**,**E2**, respectively). * *p* < 0.05, relative to control sample. Densitometry analysis was performed out of 3 scanned membranes from 3 independent experiments relative to the loading control, GAPDH.

**Figure 6 ijms-20-02670-f006:**
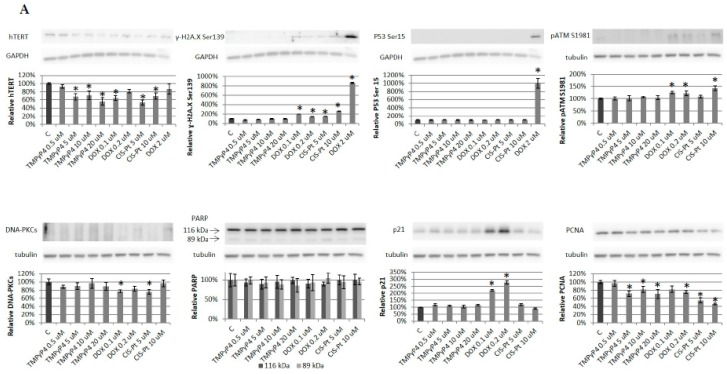
TMPyP4 does not induce DNA damage/repair-associated pathways. The influence of TMPyP4 (0.5, 5, 10 or 20 µM) on individual proteins was assessed in both MCF7 (**A**) and MDA-MB-231 cells (**B**). Cells were treated for 72 h, followed by lysis and immunodetection. Typical result out of 3 replicates was demonstrated. In experiments that were assessing proteins with a low basal level, the additional positive control was included (2 μM DOX, 2 h treatment). Densitometry analysis was performed out of 3 scanned membranes from 3 independent experiments, relative to the loading control, GAPDH (hTERT, γ-H2A.X, P53) or tubulin (pATM, DNA-PKCs, PARP, p21, PCNA). * *p* < 0.05, relative to control sample.

**Figure 7 ijms-20-02670-f007:**
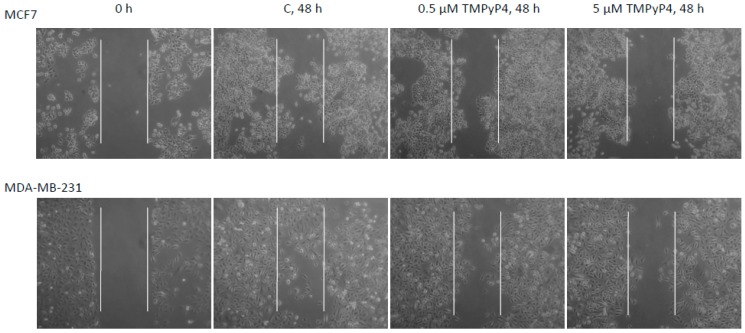
TMPyP4 triggers migration impairment in breast-cancer cells. MCF7 and MDA-MB-231 cells were incubated until reaching ca 70% confluence followed by a scratch with a sterile pipette tip. Samples were treated for 48 h with TMPyP4 (0.5 or 5 μM), and the wound healing properties were documented using a Zeiss Axiovert microscope (magnification 100 ×). Typical result out of 3 replicates was demonstrated.

**Figure 8 ijms-20-02670-f008:**
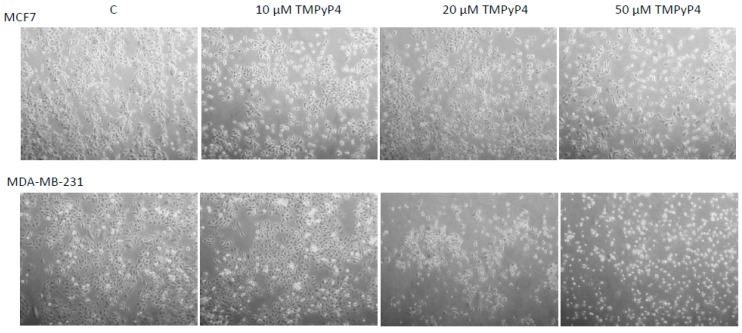
TMPyP4-mediated inhibition of adhesion properties of cancer cells. MCF7 and MDA-MB-231 cells were grown till reaching 70% confluence and exposed to 10, 20 or 50 μM TMPyP4 for 24 h. Then cells were trypsinized, and 5 × 10^5^ cells were transferred to fresh dishes. Cells were analyzed under the phase contrast microscope and photographed after 15, 30, 60 min, 3 h and 18 h (Zeiss Axiovert microscope; magnification 100×). The most significant difference was observed after 18 h. (All studied time intervals were shown in the Appendix A). Typical result out of 3 replicates was demonstrated.

**Figure 9 ijms-20-02670-f009:**
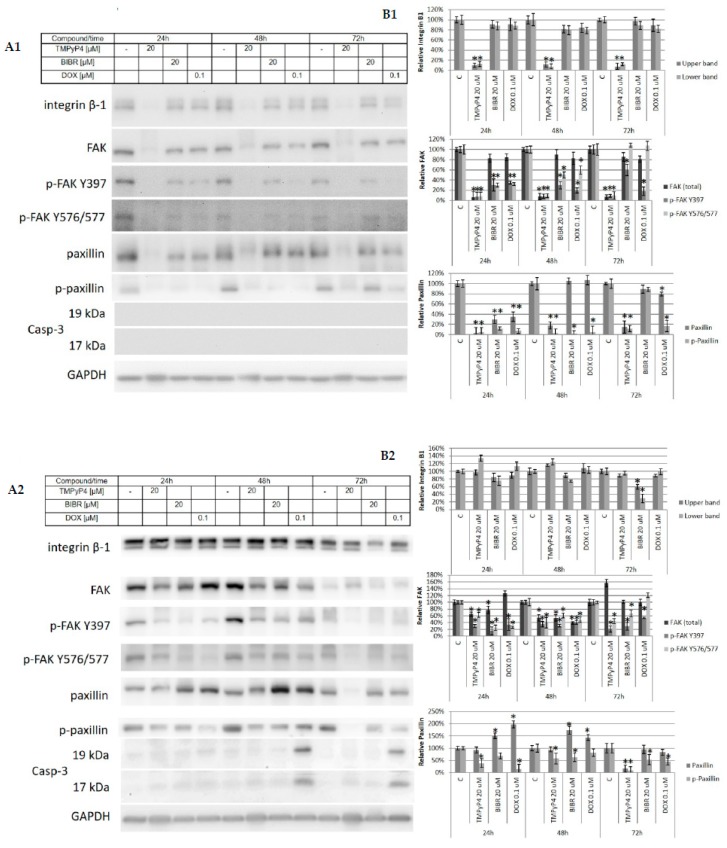
TMPyP4 down-regulates proteins engaged in adhesion pathway without inducing apoptosis. MCF7 and MDA-MB-231 cells were subjected to treatment with TMPyP4 (20 µM), BIBR 1532 (20 µM) or DOX (0.1µM) in a time-course experiment (24, 48, or 72 h). After individual time intervals, cells were lysed, and immunoidentification of B1 integrin, FAK (including p-FAK Y397 and p-FAK Y576/577) and paxillin (including p-paxillin) were performed (**A**). GAPDH was used as a loading control. Additionally, caspase-3 detection was performed, but due to a mutated caspase-3 coding gene in MCF7 cells, immunodetection did not reveal any signal. Densitometry analysis (**B**) was performed out of 3 scanned membranes from 3 independent experiments relative to the loading control, GAPDH. * *p* < 0.05, relative to control sample.

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
