# Peer review of "Telomerase Inhibitor TMPyP4 Alters Adhesion and Migration of Breast-Cancer Cells MCF7 and MDA-MB-231"

_ijms, 2019, doi:10.3390/ijms20112670_

Reviewer 1 Report

This report describes the cytotoxic properties of combining G-quadruplex binding agent TmPyP4 with the chemotherapeutic agent doxorubicin, in two breast cancer cell lines. The overall conclusion is that TmPyP4 inhibits telomerase activity through the reduction of TERT transcription, and this in turn reduced the non-canonical roles of TERT in the adhesion and migration properties of the breast cancer cells. Combining TmPyP4 treatment with doxorubicin did not result in increased cytotoxicity in the two breast cancer cell lines, regardless of molecular characteristics, through the induction of apoptosis.

Loss of viability in breast cancer models was demonstrated by a dye-based viability plate assay MTT (Figure 1, showing data from two cancer cell lines with different molecular characteristics, and the non-transformed telomerase-positive MCF12A mammary cell control), a long-term colony-forming recovery assay (Figure 4), an increase in PI-based labeling of cancer cells (Figure 5A), and an increase in the proteolytic activation of PARP (Figure 6, only positive for MDA-MB-231 cells). Cell-cycle analyses through 1) FACS of PI-labeling for DNA content or 2) western blot analyses of cell cycle phase-specific cyclins; demonstrated that combination treatment induced a pronounced G1-phase arrest in MCF7 cells and G2 arrest in MDA-MB-231 cells (Figure 5).  TmPyP4 treatments induced a pronounced loss of TERT expression at the RNA level (Figure 2) that resulted in the loss of TERT protein accumulation (Figure 3) in both breast cancer cell lines.  The authors did not detect any increase in DNA damage repair pathway activation in MCF7 cells following TmPyP4 or doxorubicin treatment, whereas the DDR marker gH2Ax was activated at baseline in MDA-MB-231 cells (Figure 6). Finally, TmPyP4 treatments resulted in the loss of migration properties in a scratch wound assay (Figure 7) and the loss of reseeding potential (Figure 8) in both cancer lines, possibly due to the reduced expression f proteins involved in the cell adhesion pathways (Figure 9).

The study is fairly straightforward, but the conclusion is marred by a lack of demonstration that TmPyP4’s specific mechanism on the observed biological pathways. Data presentations were convoluted and in many cases, the authors did not provide sufficient quantifications of their biological data, which impeded the readers’ ability to compare between treatment conditions and cell lines. While all biochemical methods for cytotoxicity, apoptosis, and cell-cycle profile measurements were adopted from classical studies or with the use of commercial kits; specific data analyses and controls were missing in some cases. This reviewer recommended the following list of mandatory revisions.  As well, several assertions made by the authors will need to be substantially clarified for the general readership. These specific points are discussed below:

1.     The authors correctly pointed out that TmPyP4’s cellular mechanisms are likely pleiotropic, attributing all its biological effects on telomerase inhibition is likely incorrect.  Specifically, while there were previous published reports on telomerase’s non-canonical functions in cancer adhesion and migration, the burden of proof that telomerase inhibition by TmPyP4 as a direct mechanism on these cancer properties has not been met.  TmPyP4 treatments induced down-regulation of proteins responsible for cancer adhesion and migration (Figure 9), but telomerase’s involvement in this regulation was not demonstrated by any empirical data, and perhaps negative demonstrated by the lack of effects in BIBR1532-treated cells.  In this context, TmPyP4 could independently affect the expression of these protein factors. It is well known that in addition to c-myc, G-quadruplex forming sequences could be found in many of the human oncogene promoters.  Conceivably, TmPyP4 treatments could stabilize these G-quadruplexes leading to altered transcription.  Have the authors considered such possibilities?

2.     Doxorubicin was added as the second combination drug at a single concentration of 0.1uM in Figures 1, 2 and 5.  What was the rationale for choosing this single concentration?  Most combination chemotherapy studies use multiple doses of both agents (See methods of Chou and Talalay, Cancer Res; 70(2); 440–6). As well, throughout the studies, western blots signals (Figures 2E, 3, 5C, 6and 9) were not quantified and normalized to the loading controls.  This makes the comparisons between treatment groups difficult and qualitative at best.  All western blots should be quantified and presented as normalized histogram combining data from all stated biological repeats.

3.     Treatment with TmPyP4 reduced cell viability (via MTT assays) by 72hr of incubation (Figure 1), without a parallel increase in apoptotic cell fractions (Figure 5, but exact definition on how the authors come up with the percent apoptotic cells were not described, most likely as the population with low PI staining/sub-G1 level; authors needed to clarify).  The authors interpreted the loss in MTT signals by the increase in G0/G1 population in MCF7 cells (G0 cell populations can also be viewed as apoptotic), and the increase in G2 arrest in MDA-MB-231 cells.  If these assertions were indeed true, then the cell cycle arrest could very well happened in the earlier time point (t=24 and 48). However, the MTT assay will not be able to differentiate between the arrested versus cycling cells (they are all viable), at the early time point, there were minimal difference in MTT readings. In the later time point (t=72) of the time course study, untreated cells accumulated increase daughter progenies while arrested cells cannot, leading to the observable differences in MTT signals.  Accordingly, the reduction of migration and adhesion properties could be interpreted by the observations that TmPyP4 treated cells are arrested at the respective cell cycle phases.  The authors should include additional data on cell cycle profiles at earlier time points (mirroring that of the migration and adhesion experiments) as well as a discussion on how this new finding could have strengthen their observations.

4.     Response to TmPyP4 cytotoxic effects, cell-cycle arrest profiles, DDR responses and migration properties, were demonstrated using two breast cancer models MCF7 and MDA-MB-231.  These two BC models have very different molecular characteristics, with MCF7 being ER+ while MDA is a triple-negative breast cancer model.  MCF7 is used as a breast cancer model with low metastatic potential in contrast to MDA-MB-231, which is readily metastatic in mouse xenograft studies. While the two models responded very differently to TmPyP4 treatments in terms of cell cycle arrest profiles and DDR engagement, they both appeared to be inhibited in the wound-healing assay. It is a missed opportunity for the authors not to speculate on the mechanism(s) behind such differences in response patterns.  The authors should include a discussion of their thoughts on these differences.

5.     Finally, there are many grammatical and stylistic errors in the text, perhaps the use of an editorial service should be recommended. A few non-exhaustive examples:

a.    Page 2, line 48, sentence is not complete.

                          b.   Page 2, line 88, agreement of tense: they are a typical model  

Author Response

Manuscript:

Telomerase inhibition with TMPyP4 alters adhesion and migration of breast cancer cells MCF7 and MDA-MB-231, by Konieczna et al.

We would like to thank the editor and reviewers for the constructive critics.

Please find enclosed response to the remarks.

1.     The authors correctly pointed out that TmPyP4’s cellular mechanisms are likely pleiotropic, attributing all its biological effects on telomerase inhibition is likely incorrect.  Specifically, while there were previous published reports on telomerase’s non-canonical functions in cancer adhesion and migration, the burden of proof that telomerase inhibition by TmPyP4 as a direct mechanism on these cancer properties has not been met. TmPyP4 treatments induced down-regulation of proteins responsible for cancer adhesion and migration (Figure 9), but telomerase’s involvement in this regulation was not demonstrated by any empirical data, and perhaps negative demonstrated by the lack of effects in BIBR1532-treated cells.  In this context, TmPyP4 could independently affect the expression of these protein factors. It is well known that in addition to c-myc, G-quadruplex forming sequences could be found in many of the human oncogene promoters.  Conceivably, TmPyP4 treatments could stabilize these G-quadruplexes leading to altered transcription.  Have the authors considered such possibilities?

 Ad 1. We do agree that a direct correlation between telomerase inhibition with TMPyP4 and adhesion or migration has not been clearly demonstrated and the issue requires further studies. One of the possible directions is the study of the potential telomere length-dependent effect but in this case the treatment must be significantly longer. If the effects are seen after 24-72 it is highly possible that it is due to non-telomeric effect. More probably associated with hTERT downregulation that was demonstrated by WB or due to general expression repression as we already mentioned in the introduction section: “Moreover, some reports suggest the complexity of TMPyP4 action in cancer cells including not only G4 stabilization but also a contribution to the regulation of expression of some genes engaged in cell metabolism, proliferation, and survival [12,13].” In fact, it is difficult to say if the repression effect results from direct interaction of TMPyP4 with promoter regions but it is possible since those region are reach in G nucleotide. It requires further studied including in silico assessment as well as CHIP experiments that we are planning to perform. This issue was extended in the discussion section.

We used BIBR 1532 to verify any potential similarity in the action of the two telomerase inhibitors effect but it appeared that this compound was not that efficient neither in hTERT downregulation (Figure 3) nor in adhesion pathway modulation (Figure 9) considering time intervals 24, 48 or 72h.

2.     Doxorubicin was added as the second combination drug at a single concentration of 0.1uM in Figures 1, 2 and 5.  What was the rationale for choosing this single concentration?  Most combination chemotherapy studies use multiple doses of both agents (See methods of Chou and Talalay, Cancer Res; 70(2); 440–6). As well, throughout the studies, western blots signals (Figures 2E, 3, 5C, 6and 9) were not quantified and normalized to the loading controls.  This makes the comparisons between treatment groups difficult and qualitative at best.  All western blots should be quantified and presented as normalized histogram combining data from all stated biological repeats.

DOX concentration i.e. 0.1 µM was chosen based on the MTT  assay (supplementary file 3). We selected the concentration that provoked the lowest significant and reproducible toxicity in order to avoid too high concentration that might reveal a nonspecific effects. On the other hand, too low concentration could show no effect due to the lack of biological potential.

And of course, several concentrations would be much better. Especially since the dose makes a huge difference and the incubation time too. However, we made a choice for the reasons presented above.

All WB were quantified (Supplementary file 1). Although they all look rather unequivocal and all alterations are confirmed by the assessment of target factors in individual pathways.

3.     Treatment with TmPyP4 reduced cell viability (via MTT assays) by 72hr of incubation (Figure 1), without a parallel increase in apoptotic cell fractions (Figure 5, but exact definition on how the authors come up with the percent apoptotic cells were not described, most likely as the population with low PI staining/sub-G1 level; authors needed to clarify). 

Yes, we did as the reviewer suspected. The addition of propidium iodide allows subsequent stages of apoptosis and eventual cell death to be distinguished [Duensing TD, Watson SR. Assessment of Apoptosis (Programmed Cell Death) by Flow Cytometry. Cold Spring Harb Protoc. 2018 Jan 2;2018(1). doi: 10.1101/pdb.prot093807]. Consequently, in the presented study, cells with reduced PI staining of DNA (sub-G1 fraction) were considered apoptotic. Even if not very precise, this method enabled quick verification of the apoptotic population.

The authors interpreted the loss in MTT signals by the increase in G0/G1 population in MCF7 cells (G0 cell populations can also be viewed as apoptotic), and the increase in G2 arrest in MDA-MB-231 cells.  If these assertions were indeed true, then the cell cycle arrest could very well happened in the earlier time point (t=24 and 48). However, the MTT assay will not be able to differentiate between the arrested versus cycling cells (they are all viable), at the early time point, there were minimal difference in MTT readings. In the later time point (t=72) of the time course study, untreated cells accumulated increase daughter progenies while arrested cells cannot, leading to the observable differences in MTT signals.  Accordingly, the reduction of migration and adhesion properties could be interpreted by the observations that TmPyP4 treated cells are arrested at the respective cell cycle phases.  The authors should include additional data on cell cycle profiles at earlier time points (mirroring that of the migration and adhesion experiments) as well as a discussion on how this new finding could have strengthen their observations.

Since apoptosis was expected, caspase-3 activation was assessed in MDA-MB-231 but no alteration was reported. In MCF7 (caspase-3 negative) it was impossible to assess caspase-3, due to issues mentioned in the Results section. For this reason cell cycle assessment was performed in 24 and 48 h, revealing no significant alterations, including no apoptosis (see Supplementary file 3).

4.     Response to TmPyP4 cytotoxic effects, cell-cycle arrest profiles, DDR responses and migration properties, were demonstrated using two breast cancer models MCF7 and MDA-MB-231.  These two BC models have very different molecular characteristics, with MCF7 being ER+ while MDA is a triple-negative breast cancer model.  MCF7 is used as a breast cancer model with low metastatic potential in contrast to MDA-MB-231, which is readily metastatic in mouse xenograft studies. While the two models responded very differently to TmPyP4 treatments in terms of cell cycle arrest profiles and DDR engagement, they both appeared to be inhibited in the wound-healing assay. It is a missed opportunity for the authors not to speculate on the mechanism(s) behind such differences in response patterns.  The authors should include a discussion of their thoughts on these differences.

In fact, both studied cell lines are different as described in the text. Noteworthy, concerning the DDR assessment, MCF7 revealed a very low basal level of phospho p53 as well as γH2A.X while both proteins were highly elevated in control MDA-MB-231 cells. However, no DDR was induced in none of the cell lines after TMPyP4 treatment. Concerning the cell cycle assessment – the effects were different in both cell lines in a semi quantitative WB analysis but in fact the trend was similar. Altogether it might be due to different resistance/sensitivity level to the compound resulting from different signaling in both cell lines. Altogether, MDA-MB-231 cells were more resistant in MTT and, similarly, the cell cycle of those cells was only moderately affected when TMPyP4 was involved. It was commented in the discussion section.

 5.     Finally, there are many grammatical and stylistic errors in the text, perhaps the use of an editorial service should be recommended. A few non-exhaustive examples:

a.    Page 2, line 48, sentence is not complete.

                          b.   Page 2, line 88, agreement of tense: they are a typical model  

The manuscript was thoroughly revised.

Reviewer 2 Report

- The authors carried out a large amount of work and planned the experiment well. According to the author, the aim of the work is to study the mechanism of action of the telomerase inhibitor TMPyP4. The topic of research is relevant and causes interest to the reader.
- It is necessary to describe in more detail the need to use another inhibitor in the experiment "BIBR". It is required to add the substantiation and the principle of action of the substances/inhibitors used for different experiments.
- In the discussion chapter, it is necessary to insert links to all the Figure (# 3, # 8, # 9).
- What is the basis of the choice of different concentrations of  TMPyP4 and other substances for each unit of research? Why were some concentrations tested depending on the experiment?
- The cell line with a mutation in the gene Caspase-3, but the authors do not explain this, in fact, it can be seen only from the results of the western blot. It is necessary to insert a reference to the cell line passport for this statement: "Note on the MCF7 cells". We need to better justify the choice of cell line MCF-12A.

- Figure 8. "... photographed after 15, 30, 60 min, 3 h and 18 h (Zeiss Axiovert microscope; magnification 100x). The most significant difference was observed after 18 h. There are no pictures at other hours.” The shows are different time intervals, but the images are presented only for 18 hours. Justify why different time intervals are used in Figure 7 and Figure 8 is not clear from the text of the article. 

-Conclusions need to be detailed and concretized. The authors of the work have done extensive work with a wide range of methods for evaluating different molecular mechanisms, but it is not clear from the conclusion that the authors wanted to show in the work. The phrase "various molecular mechanisms" is not accurate.

Author Response

Manuscript:

Telomerase inhibition with TMPyP4 alters adhesion and migration of breast cancer cells MCF7 and MDA-MB-231, by Konieczna et al.

We would like to thank the editor and reviewers for the constructive critics.

Please find enclosed response to the remarks.

Comments and Suggestions for Authors

- The authors carried out a large amount of work and planned the experiment well. According to the author, the aim of the work is to study the mechanism of action of the telomerase inhibitor TMPyP4. The topic of research is relevant and causes interest to the reader.
- It is necessary to describe in more detail the need to use another inhibitor in the experiment "BIBR". It is required to add the substantiation and the principle of action of the substances/inhibitors used for different experiments.

We used BIBR 1532 to verify any potential similarity in the action of the two telomerase inhibitors effect but it appeared that this compound was not that efficient neither in hTERT downregulation (Figure 3) nor in adhesion pathway modulation (Figure 9) considering time intervals 24, 48 or 72h.
- In the discussion chapter, it is necessary to insert links to all the Figure (# 3, # 8, # 9).

It is done now.
- What is the basis of the choice of different concentrations of  TMPyP4 and other substances for each unit of research? Why were some concentrations tested depending on the experiment?

We were looking for sensitivity level of cells exposed to TMPyP4. For this reason in the individual assays the lowest possible concentration that revealed an effect was used to avoid triggering side, non-specific effects. TMPyP4 was generally chosen on the base of hTERT modulation ability. 20µM was representing the lowest to show significant downregulation of the telomerase catalytic subunit (Figure 6) without a dramatic loss of cell viability (MTT assay, Figure 1). Thus, it was used to assess most metabolic pathways.

A broader range of concentration (0.5 to 20uM) was used to assess for DDR-associated pathway to evaluate possible sensitivity level. Similarly, lower concentrations were used in the scratch assay to eliminate toxicity effect.

- The cell line with a mutation in the gene Caspase-3, but the authors do not explain this, in fact, it can be seen only from the results of the western blot. It is necessary to insert a reference to the cell line passport for this statement: "Note on the MCF7 cells". We need to better justify the choice of cell line MCF-12A.

MCF7 cells are caspase-3 deficient that determines the resistance of those cells to some cytotoxic agents [Yang S, Zhou Q, Yang X. Caspase-3 status is a determinant of the differential responses to genistein between MDA-MB-231 and MCF-7 breast cancer cells. Biochim Biophys Acta 2007, 1773 (6), 903-911].

Additionally, a non-tumorigenic epithelial cell line, MCF-12A, was also used as a reference cell line that is reported to show residual telomerase expression and activity. Importantly, it is derived from the same type of tissue as studied cancer cell lines.

- Figure 8. "... photographed after 15, 30, 60 min, 3 h and 18 h (Zeiss Axiovert microscope; magnification 100x). The most significant difference was observed after 18 h. There are no pictures at other hours.” The shows are different time intervals, but the images are presented only for 18 hours. Justify why different time intervals are used in Figure 7 and Figure 8 is not clear from the text of the article. 

All the missing time points are included in the Supplementary file 2.

Those two experiments (migration/scratch and adhesion) are made using a different protocol. In the scratch assay, 24 h treatment did not show any significant results and for this reason the treatment was longer (48 h).

-Conclusions need to be detailed and concretized. The authors of the work have done extensive work with a wide range of methods for evaluating different molecular mechanisms, but it is not clear from the conclusion that the authors wanted to show in the work. The phrase "various molecular mechanisms" is not accurate.

Conclusions were amended.

Reviewer 3 Report

This manuscript aims to elucidate the anticancer properties of TMPyP4 in breast cancer cells. The authors examine the cytotoxic effects, cell cycle alterations, DNA damage and clonogenic properties of BRCA cells after treated with TMPyP4. The results showed the non-canonical anti-cancer effects TMPyP4 in BRCA cells via downregulating proteins engaged in adhesion pathway and triggering migration impairment

Specific suggestions about the manuscript are indicated below.

1. Figure 1: the statistical difference only shown in MCF7 treated with TMPyP4 for 48 hrs. I suggest the statistical symbols should also be presented in Figure 1 in the groups with significant decreased in cell survival compared to control groups.

2.  In line 144, BIBR 1532 was firstly mentioned, please explain the function of BIBR 1532, and the rationale that BIBR 1532 used in this study.

3. Figure 5, the result of flow cytometry showed different patterns of MCF-7 (G1 arrest) and MDA-MB-231 (S and G2/M arrest) after treatment. However, similar results of cyclin expression were observed by Western blot analysis. How do you explain these mechanisms?

4. The possible mechanisms of how TMPyP4 downregulates proteins engaged in adhesion pathway and triggers migration impairment in BRCA cells should be discussed more detail.

5. The present study examined several pathways but lacked of analysis profoundly. 

Author Response

Manuscript:

Telomerase inhibition with TMPyP4 alters adhesion and migration of breast cancer cells MCF7 and MDA-MB-231, by Konieczna et al.

We would like to thank the editor and reviewers for the constructive critics.

Please find enclosed response to the remarks.

1. Figure 1: the statistical difference only shown in MCF7 treated with TMPyP4 for 48 hrs. I suggest the statistical symbols should also be presented in Figure 1 in the groups with significant decreased in cell survival compared to control groups.

It is shown now.

2.  In line 144, BIBR 1532 was firstly mentioned, please explain the function of BIBR 1532, and the rationale that BIBR 1532 used in this study.

We used BIBR 1532 to verify any potential similarity in the action of the two telomerase inhibitors effect but it appeared that this compound was not that efficient neither in hTERT downregulation (Figure 3) nor in adhesion pathway modulation (Figure 9) considering time intervals 24, 48 or 72h.

3. Figure 5, the result of flow cytometry showed different patterns of MCF-7 (G1 arrest) and MDA-MB-231 (S and G2/M arrest) after treatment. However, similar results of cyclin expression were observed by Western blot analysis. How do you explain these mechanisms?

Concerning the cell cycle assessment – the effects were different in both cell lines in a semi quantitative WB analysis but in fact the trend was similar. Altogether it might be due to different resistance/sensitivity level to the compound resulting from different signaling in both cell lines. Altogether, MDA-MB-231 cells were more resistant in MTT and, similarly, the cell cycle of those cells was only moderately affected when TMPyP4 was involved.

We do agree that a direct correlation between telomerase inhibition with TMPyP4 and cell cycle has not been clearly demonstrated and the issue requires further studies. One of the possible directions is the study of the potential telomere length-dependent effect but in this case the treatment must be significantly longer. If the effects are seen after 24-72 it is highly possible that it is due to non-telomeric effect. More probably associated with hTERT downregulation that was demonstrated by WB or due to general expression repression as we already mentioned in the introduction section: “Moreover, some reports suggest the complexity of TMPyP4 action in cancer cells including not only G4 stabilization but also a contribution to the regulation of expression of some genes engaged in cell metabolism, proliferation, and survival [12,13].” In fact, it is difficult to say if the repression effect results from direct interaction of TMPyP4 with promoter regions but it is possible since those region are reach in G nucleotide. It requires further studied including in silico assessment as well as CHIP experiments that we are planning to perform. This issue was extended in the discussion section.

4. The possible mechanisms of how TMPyP4 downregulates proteins engaged in adhesion pathway and triggers migration impairment in BRCA cells should be discussed more detail.

It is discussed now.

5. The present study examined several pathways but lacked of analysis profoundly. 

Discussion is now deeper.

Round  2

Reviewer 1 Report

In my previous review, I asked the authors to provide more quantitative data to substantiate their claims, as well as a more thorough discussion on the pleothora of TmPyP4’s mechanisms.  While the authors satisfactorily answered some of these queries, there are still substantial logical gaps and stylistic errors that need to be addressed.  These points are listed below:

1.     The authors correctly pointed out that TmPyP4’s cellular mechanisms are likely pleiotropic, attributing all its biological effects on telomerase inhibition is likely incorrect.  Specifically, while there were previous published reports on telomerase’s non-canonical functions in cancer adhesion and migration, the burden of proof that telomerase inhibition by TmPyP4 as a direct mechanism on these cancer properties has not been met.  TmPyP4 treatments induced down-regulation of proteins responsible for cancer adhesion and migration (Figure 9), but telomerase’s involvement in this regulation was not demonstrated by any empirical data, and perhaps negative demonstrated by the lack of effects in BIBR1532-treated cells.  In this context, TmPyP4 could independently affect the expression of these protein factors. It is well known that in addition to c-myc, G-quadruplex forming sequences could be found in many of the human oncogene promoters.  Conceivably, TmPyP4 treatments could stabilize these G-quadruplexes leading to altered transcription.  Have the authors considered such possibilities?

Ad 1. We do agree that a direct correlation between telomerase inhibition with TMPyP4 and adhesion or migration has not been clearly demonstrated and the issue requires further studies. One of the possible directions is the study of the potential telomere length-dependent effect but in this case the treatment must be significantly longer. If the effects are seen after 24-72 it is highly possible that it is due to non-telomeric effect. More probably associated with hTERT downregulation that was demonstrated by WB or due to general expression repression as we already mentioned in the introduction section: “Moreover, some reports suggest the complexity of TMPyP4 action in cancer cells including not only G4 stabilization but also a contribution to the regulation of expression of some genes engaged in cell metabolism, proliferation, and survival [12,13].” In fact, it is difficult to say if the repression effect results from direct interaction of TMPyP4 with promoter regions but it is possible since those region are reach in G nucleotide. It requires further studied including in silico assessment as well as CHIP experiments that we are planning to perform. This issue was extended in the discussion section. We used BIBR 1532 to verify any potential similarity in the action of the two telomerase inhibitors effect but it appeared that this compound was not that efficient neither in hTERT downregulation (Figure 3) nor in adhesion pathway modulation (Figure 9) considering time intervals 24, 48 or 72h.

While this reviewer is pleased that the authors agreed TmPyP4 could have biological effects independent of telomerase activity/transcription modulations, and is planning to address the promoter binding sequence of TmPyP4 with ChIP experiments, the current discussion still focused on TmPyP4’s effects on TERT as the only explanation of the changes in adhesion and migration properties of the treated cells.  This discrepancy need to be addressed to avoid misleading the readers that the mechanism has been solved.

2.     Doxorubicin was added as the second combination drug at a single concentration of 0.1uM in Figures 1, 2 and 5.  What was the rationale for choosing this single concentration?  Most combination chemotherapy studies use multiple doses of both agents (See methods of Chou and Talalay, Cancer Res; 70(2); 440–6). As well, throughout the studies, western blots signals (Figures 2E, 3, 5C, 6and 9) were not quantified and normalized to the loading controls.  This makes the comparisons between treatment groups difficult and qualitative at best.  All western blots should be quantified and presented as normalized histogram combining data from all stated biological repeats.

DOX concentration i.e. 0.1 µM was chosen based on the MTT  assay (supplementary file 3). We selected the concentration that provoked the lowest significant and reproducible toxicity in order to avoid too high concentration that might reveal a nonspecific effects. On the other hand, too low concentration could show no effect due to the lack of biological potential. And of course, several concentrations would be much better. Especially since the dose makes a huge difference and the incubation time too. However, we made a choice for the reasons presented above. All WB were quantified (Supplementary file 1). Although they all look rather unequivocal and all alterations are confirmed by the assessment of target factors in individual pathways.

Thank you for providing the quantification data. They should be included in the main figures and not as supplemental information.  As well, legends are missing for these quantification data.  How many blots were quantified? Are they biological or technical repeats? These are important information to be included.

3.     Treatment with TmPyP4 reduced cell viability (via MTT assays) by 72hr of incubation (Figure 1), without a parallel increase in apoptotic cell fractions (Figure 5, but exact definition on how the authors come up with the percent apoptotic cells were not described, most likely as the population with low PI staining/sub-G1 level; authors needed to clarify).  The authors interpreted the loss in MTT signals by the increase in G0/G1 population in MCF7 cells (G0 cell populations can also be viewed as apoptotic), and the increase in G2 arrest in MDA-MB-231 cells.  If these assertions were indeed true, then the cell cycle arrest could very well happened in the earlier time point (t=24 and 48). However, the MTT assay will not be able to differentiate between the arrested versus cycling cells (they are all viable), at the early time point, there were minimal difference in MTT readings. In the later time point (t=72) of the time course study, untreated cells accumulated increase daughter progenies while arrested cells cannot, leading to the observable differences in MTT signals.  Accordingly, the reduction of migration and adhesion properties could be interpreted by the observations that TmPyP4 treated cells are arrested at the respective cell cycle phases.  The authors should include additional data on cell cycle profiles at earlier time points (mirroring that of the migration and adhesion experiments) as well as a discussion on how this new finding could have strengthen their observations.

Yes, we did as the reviewer suspected. The addition of propidium iodide allows subsequent stages of apoptosis and eventual cell death to be distinguished [Duensing TD, Watson SR. Assessment of Apoptosis (Programmed Cell Death) by Flow Cytometry. Cold Spring Harb Protoc. 2018 Jan 2;2018(1). doi: 10.1101/pdb.prot093807]. Consequently, in the presented study, cells with reduced PI staining of DNA (sub-G1 fraction) were considered apoptotic. Even if not very precise, this method enabled quick verification of the apoptotic population.

The author did not address the second part of my question, which is whether the cell cycle-arrest phenotypes could explain the differences between MTT versus CFU viability data.  I asked specifically for the additional analyses of cell cycle profiles at earlier time points. This should be done for a more concrete conclusion of the cell viability versus cell cycle exit phenotype.

4.     Response to TmPyP4 cytotoxic effects, cell-cycle arrest profiles, DDR responses and migration properties, were demonstrated using two breast cancer models MCF7 and MDA-MB-231.  These two BC models have very different molecular characteristics, with MCF7 being ER+ while MDA is a triple-negative breast cancer model.  MCF7 is used as a breast cancer model with low metastatic potential in contrast to MDA-MB-231, which is readily metastatic in mouse xenograft studies. While the two models responded very differently to TmPyP4 treatments in terms of cell cycle arrest profiles and DDR engagement, they both appeared to be inhibited in the wound-healing assay. It is a missed opportunity for the authors not to speculate on the mechanism(s) behind such differences in response patterns.  The authors should include a discussion of their thoughts on these differences.

In fact, both studied cell lines are different as described in the text. Noteworthy, concerning the DDR assessment, MCF7 revealed a very low basal level of phospho p53 as well as γH2A.X while both proteins were highly elevated in control MDA-MB-231 cells. However, no DDR was induced in none of the cell lines after TMPyP4 treatment. Concerning the cell cycle assessment – the effects were different in both cell lines in a semi quantitative WB analysis but in fact the trend was similar. Altogether it might be due to different resistance/sensitivity level to the compound resulting from different signaling in both cell lines. Altogether, MDA-MB-231 cells were more resistant in MTT and, similarly, the cell cycle of those cells was only moderately affected when TMPyP4 was involved. It was commented in the discussion section.

Precisely my point: that the two breast cancer models have very different signaling dependency and threshold DDR activation levels. It may be necessary to tune down the argument that TmPyP4 treatments are causing the same effects in the two models. In fact, even when one only consider the anti-adhesion/migration effects of TmPyP4, the two cell lines responded differently: Integrin 1b expression was inhibited in MCF7 but not MDA-MB-231.

5.     Finally, there are many grammatical and stylistic errors in the text, perhaps the use of an editorial service should be recommended. A few non-exhaustive examples:

a.    Page 2, line 48, sentence is not complete.

b.    Page 2, line 88, agreement of tense: they are a typical model

Unfortunately, many of the same mistakes and logical loops were not corrected.  The authors may consider professional editorial service for these tasks.

Author Response

Manuscript:

Telomerase inhibitor TMPyP4 alters adhesion and migration of breast cancer cells MCF7 and MDA-MB-231, by Konieczna et al.

We would like to thank the editor and reviewer for the constructive critics.

Please find enclosed response to the remarks, labeled with blue color.

Importantly, the title of the manuscript was slightly modified.

Comments and Suggestions for Authors

In my previous review, I asked the authors to provide more quantitative data to substantiate their claims, as well as a more thorough discussion on the pleothora of TmPyP4’s mechanisms.  While the authors satisfactorily answered some of these queries, there are still substantial logical gaps and stylistic errors that need to be addressed.  

While this reviewer is pleased that the authors agreed TmPyP4 could have biological effects independent of telomerase activity/transcription modulations, and is planning to address the promoter binding sequence of TmPyP4 with ChIP experiments, the current discussion still focused on TmPyP4’s effects on TERT as the only explanation of the changes in adhesion and migration properties of the treated cells.  This discrepancy need to be addressed to avoid misleading the readers that the mechanism has been solved.

We did alter discussion accordingly and underlined the pleiotropic effect of the porphyrin.

2.     Thank you for providing the quantification data. They should be included in the main figures and not as supplemental information.  As well, legends are missing for these quantification data.  How many blots were quantified? Are they biological or technical repeats? These are important information to be included.

All the supporting materials are included in the main body of the text now.

Legends were supplemented with critical data – MTT was performed in triplicates (biological replicates) and each experiment consisted of 8 samples/wells.

Concerning western blot - briefly, all experiments were performed in triplicates at least and densitometry analysis was performed from all the membranes.

3.     The author did not address the second part of my question, which is whether the cell cycle-arrest phenotypes could explain the differences between MTT versus CFU viability data.  I asked specifically for the additional analyses of cell cycle profiles at earlier time points. This should be done for a more concrete conclusion of the cell viability versus cell cycle exit phenotype.

Yes, the cell arrest phenotype in MCF7 cells may explain the differences between MTT versus CFU viability data. Especially since clonogenic reflects a bit different mechanisms. Additionally, in this assay cells are dispersed and consequently are more sensitive to potentially toxic compounds.

And again, the cycle profiles at earlier time points in MCF7 or caspase-3 assessment in MDA-MB-231 did not show any significant alterations in viability, arrest or apoptosis.

4.     Precisely my point: that the two breast cancer models have very different signaling dependency and threshold DDR activation levels. It may be necessary to tune down the argument that TmPyP4 treatments are causing the same effects in the two models. In fact, even when one only consider the anti-adhesion/migration effects of TmPyP4, the two cell lines responded differently: Integrin 1b expression was inhibited in MCF7 but not MDA-MB-231.

Yes, tuning down the statement may be the best option, thank you for the suggestion.

A different response of both studied cells may be explained by a trivial statement that different cells respond in a different way. It would be a superficial but might be a very true at the same time. However, we may speculate that observed differences in the response to the studied compound may result from some very particular conditions. It might result from the fact that both cell lines reveal different basal resistance to drugs that is very often demonstrated in numerous papers. MDA-MB-231 cells are usually claimed to be more resistant to chemotherapeutic drugs e.g. doxorubicin (also confirmed in our studies, Fig. 1). On the other hand, MCF7 cells reveal significantly higher basal level of ABCB1. Thus, we did try to identify any potential association of the cell survival response with the level of ABCB1 expression or activity but we could not find any influence of TMPyP4 on none of those mechanisms (data not shown). However, even if it does not inhibit ABCB1, it does not mean they it is not transported by the pump. Generally, bioavailability might be an issue. We cannot tell yet what is the mechanism that leads to some interactions that provoke a different response of those two different cell lines to the same compound.

5.     Finally, there are many grammatical and stylistic errors in the text, perhaps the use of an editorial service should be recommended. A few non-exhaustive examples:

a.    Page 2, line 48, sentence is not complete.

b.    Page 2, line 88, agreement of tense: they are a typical model

 Unfortunately, many of the same mistakes and logical loops were not corrected.  The authors may consider professional editorial service for these tasks.

It was corrected.

Reviewer 3 Report

I have no more questions.

Author Response

Thank you for your kind comments.

Round  3

Reviewer 1 Report

In my previous reviews, I asked the authors to provide more quantitative data to substantiate their claims, as well as a more thorough discussion on the plethora of TmPyP4’s mechanisms. In their second revision, many of these concerns were addressed.  However, the presentation of data and the discussion (particularly the new conclusion) could use another round of polish. These points are listed below:

1. Quantification data for the western blots are now included.  However, the presentations of these data are not in a way that is easily assessable by the reader.  The authors may consider rearranging the graphics of the figures such that the quantification data will be immediately juxtaposition to the corresponding western blots.

2. While this reviewer appreciate the authors’ effort in tuning down the direct telomerase inhibition effects of TmPyP4 on cell cycle and migration effects, the new conclusion prose does not read well, and is not easily understandable.  This problem may be easily addressed with the help of a professional editing service.

Author Response

Manuscript:

Telomerase inhibitor TMPyP4 alters adhesion and migration of breast cancer cells MCF7 and MDA-MB-231, by Konieczna et al.

We would like to thank the editor and reviewer for the constructive critics.

Please find enclosed response to the remarks – all amendments in the text are labeled with yellow color.

Comments and Suggestions for Authors

In my previous reviews, I asked the authors to provide more quantitative data to substantiate their claims, as well as a more thorough discussion on the plethora of TmPyP4’s mechanisms. In their second revision, many of these concerns were addressed.  However, the presentation of data and the discussion (particularly the new conclusion) could use another round of polish. These points are listed below:

1.     Quantification data for the western blots are now included.  However, the presentations of these data are not in a way that is easily assessable by the reader.  The authors may consider rearranging the graphics of the figures such that the quantification data will be immediately juxtaposition to the corresponding western blots.

It was corrected

2. While this reviewer appreciate the authors’ effort in tuning down the direct telomerase inhibition effects of TmPyP4 on cell cycle and migration effects, the new conclusion prose does not read well, and is not easily understandable. This problem may be easily addressed with the help of a professional editing service.

The text was corrected by a native speaker.